# Measuring the optimal exposure for single particle cryo-EM using a 2.6 Å reconstruction of rotavirus VP6

**Timothy Grant[1], Nikolaus Grigorieff[1,2]***

[1]Janelia Research Campus, Howard Hughes Medical Institute, Ashburn, United States; [2]Department of Biochemistry, Rosenstiel Basic Medical Sciences Research Center, Brandeis University, Waltham, United States

**Abstract** Biological specimens suffer radiation damage when imaged in an electron microscope, ultimately limiting the attainable resolution. At a given resolution, an optimal exposure can be defined that maximizes the signal-to-noise ratio in the image. Using a 2.6 Å resolution single particle cryo-EM reconstruction of rotavirus VP6, determined from movies recorded with a total exposure of 100 electrons/$Å^2$, we obtained accurate measurements of optimal exposure values over a wide range of resolutions. At low and intermediate resolutions, our measured values are considerably higher than obtained previously for crystalline specimens, indicating that both images and movies should be collected with higher exposures than are generally used. We demonstrate a method of using our optimal exposure values to filter movie frames, yielding images with improved contrast that lead to higher resolution reconstructions. This 'high-exposure' technique should benefit cryo-EM work on all types of samples, especially those of relatively low-molecular mass.

*For correspondence: niko@ grigorieff.org

**Competing interests:** The authors declare that no competing interests exist.

## Introduction

Electron microscopy of isolated macromolecules and their complexes (single particles) embedded in a thin layer of vitreous ice (cryo-EM) has recently led to a number of structures determined at near-atomic resolution (*Liao et al., 2013*; *Allegretti et al., 2014*; *Bartesaghi et al., 2014*; *Wong et al., 2014*), a level of detail that had previously been restricted to X-ray crystallography and NMR (for a recent review of the technique, see *Cheng et al., 2015*). A limiting factor in the resolution of EM images of biological specimens is radiation damage because the imaging of such specimens ultimately relies on the interaction of electrons with the sample. Some of these interactions will result in energy being deposited in the specimen, and these will cause radiation damage (*Glaeser, 1971*; *Henderson, 1995*). The radiation damage fundamentally limits the information present in the images because the useful signal added per incident electron decreases with increasing electron exposure, while the added noise (image contrast originating from other parts of the sample as well as inelastic scattering) remains approximately constant. If the signal gain per unit exposure is known at a given resolution, an optimal exposure can be chosen that will maximize the signal-to-noise ratio (SNR) at that resolution (*Hayward and Glaeser, 1979*; *Baker et al., 2010*).

The rate of exposure-dependent signal decay has been measured by following the intensities of the fading diffraction spots in exposure series obtained from 2D and thin 3D crystals (*Unwin and Henderson, 1975*; *Hayward and Glaeser, 1979*; *Stark et al., 1996*; *Baker et al., 2010*). These studies demonstrated that higher resolution intensities tend to fade faster than lower resolution intensities, and that the rate of fading for all resolutions is slowed under liquid nitrogen conditions relative to room temperature conditions. The fading of the spots can be described by an exponential

**eLife digest** Microscopes allow us to visualize objects that are invisible to the naked eye. One type of microscope—called the electron microscope—produces images using beams of particles known as electrons, which enables them to produce more detailed images than microscopes that use light.

There are several ways to prepare samples for electron microscopy. For example, in 'electron cryo-microscopy'—or cryo-EM for short—a sample is rapidly frozen to preserve its features before it is examined under the microscope. This technique generates images that can be analyzed by computers to produce three-dimensional models of individual viruses, proteins, and other tiny objects. Unfortunately, the samples need to be exposed to high-energy beams of electrons that will damage the sample while the images are gathered, which results in sample movement and blurry images that lack the finer details.

The contrast between the sample and its background is one of the factors that determine the final quality of an image. The higher the contrast, the greater the level of structural information that can be obtained, but this requires the use of longer exposures to the electron beam. To overcome this issue, researchers found that instead of recording a single image, it is possible to record movies in which the movement of the sample under the electron beam can be tracked. After the movies are gathered, the movie frames are aligned using computer software to reduce the blurring caused by the sample moving and can then be used to make three-dimensional models.

Grant and Grigorieff improved this method further by studying how quickly a large virus-like particle called 'rotavirus double-layered particle' is damaged under the electron beam. These experiments identified an optimum range of exposure to electrons that provides the highest image contrast at any given level of detail. These findings were used to design an exposure filter that can be applied to the movie frames, allowing Grant and Grigorieff to visualize features of the virus that had not previously been observed by cryo-EM.

This method was also used to study an assembly of proteins known as the proteasome, which is responsible for destroying old proteins. Grant and Grigorieff's findings should be useful for cryo-EM studies on many kinds of samples.

decay that is characterized by $N_e$, the resolution-dependent critical exposure after which spot intensities are reduced to 1/e of their starting values. To perform similar experiments with non-crystalline material, that is single particles, the resolution-dependent fading of the signal present in an image (or average of images) has to be determined. Unlike diffraction spots obtained from crystals, the signal in an image will give rise to signal amplitudes in the image Fourier transform. The squared Fourier amplitudes are equivalent to crystal diffraction intensities but will be affected by a number of additional factors, for example, the contrast transfer function (CTF) of the microscope and movement of the specimen, both attenuating the amplitudes. However, assuming a linear imaging system and constant amplitude-attenuating factors throughout an exposure series, the rate of fading of the signal in an image can be determined from the fading of the signal amplitudes in the image Fourier transform. While the fading of spots can be measured directly from diffraction patterns, the signal amplitudes calculated from an image are superimposed on the noise generated by the stochastic detection of electrons forming the image, as well as background due to the embedding ice and support film if present. Therefore, instead of measuring signal amplitudes directly, they have to be estimated from measurements of the SNR. Since the exposure, and therefore the noise, in each image within a series remains approximately constant, the SNR is directly proportional to the signal in each image. Analogous to the fading of diffraction spots (*Hayward and Glaeser, 1979*), the SNR of an image recorded after a prior exposure of $N$ electrons/Å$^2$ is therefore

$$\mathrm{SNR}(k, N) = \mathrm{SNR}(k, 0)\, e^{-\frac{N}{N_e(k)}}, \qquad (1)$$

where $k$ is the spatial frequency, $N$ is the accumulated exposure, and $N_e$ is the resolution-dependent critical exposure defined above. We can see from *Equation 1* that a plot of $\ln(\mathrm{SNR}(k))$ vs $N$ should have a slope of $-1/N_e(k)$, and *Baker et al. (2010)* demonstrated that this relationship is true not just in the case of measurements of instantaneous signal, but also in the case of a detector that

integrates measured intensities over time. The SNR of an image will continue to increase with electron exposures larger than the critical exposure. *Hayward and Glaeser (1979)* showed that the signal present in crystal diffraction spots is maximized at an optimal exposure of ~2.5 times the critical exposure. The SNR increase per unit exposure is larger below the optimal exposure than the corresponding rate of SNR decay beyond the optimal exposure; thus, it is typically better to slightly overexpose a specimen than slightly underexpose. An exposure of 2.5 $N_e$ is also expected to optimize the signal in images at a given resolution due to the analogy between image and crystal data as explained above.

Despite studies of the general effects of radiation damage on single-particle specimens (*Conway et al., 1993*), direct measurement of the critical exposure on non-crystalline specimens has proven difficult in the past for a number of reasons. Firstly, recording a series of images with increasing exposure would have led to images with SNRs so low that they would be difficult to analyze. Secondly, until recently the resolutions commonly obtained with the single-particle method were of insufficient quality to follow radiation damage at near-atomic resolution (better than 4 Å). Finally, signal loss due to beam-induced specimen movement (*Brilot et al., 2012*; *Campbell et al., 2012*; *Li et al., 2013*; *Scheres, 2014*) was hard to quantify, and thus, the effects of radiation damage and movement could not be separated. The recent availability of direct electron detectors (*Milazzo et al., 2005*; *Faruqi and Henderson, 2007*; *Li et al., 2013*) has largely alleviated these problems. The ability to record movies instead of single images allows for easy collection of a continuous exposure series, while allowing for measurement and correction of specimen movement (*Brilot et al., 2012*; *Campbell et al., 2012*; *Li et al., 2013*; *Scheres, 2014*). Furthermore, the improved detective quantum efficiency of the detectors (*Ruskin et al., 2013*; *McMullan et al., 2014*) has enhanced the signal present in the images, and this enhancement combined with motion correction has enabled single-particle reconstructions to reach higher resolutions. In this study, we used a K2 Summit direct detector (Gatan, Inc., Pleasanton, CA) to collect a single particle data set of rotavirus double-layered particle (DLP) and estimate the critical exposure of a non-crystalline specimen. Rotavirus DLP is a spherical particle with a diameter of about 700 Å and a molecular mass of about 70 MDa (*Grigorieff and Harrison, 2011*). The capsid has icosahedral symmetry and consists predominantly of viral proteins VP2 and VP6, forming an inner and outer layer, respectively. They have a combined mass of about 44 MDa and package 11 or 12 copies of viral proteins VP1 and VP3, as well as 11 dsRNA segments that make up the viral genome (*Estrozi et al., 2013*). VP6 assembles into 260 trimers that arrange in a $T = 13$ surface lattice containing a total of 780 VP6 monomers. The $T = 13$ lattice enables 13-fold averaging of density in addition to the 60-fold icosahedral symmetry. The large size of DLP and its high symmetry make it an ideal specimen for this study, yielding high-contrast particles with very low-alignment errors (*Henderson et al., 2011*) and almost a thousand averaged subunits per particle.

The DLP data presented here are of sufficient quality to obtain sub-3-Å resolution reconstructions using just a few movie frames, enabling us to accurately measure the critical exposure over a large range of resolutions. To test the new filtering scheme, we also show its application to a recently published reconstruction of the *Thermoplasma acidophilum* 20S proteasome at 2.8 Å resolution (*Campbell et al., 2015*), demonstrating optimization of SNR in the final reconstruction and improved alignment when the SNR in the images is very low.

## Results

### 2.6 Å reconstruction of rotavirus DLP

Examples of an aligned movie and particle images with different effective exposures are shown in *Figure 1*. Processing exposure-filtered images (see below) of ~4000 rotavirus particles and exploiting both their icosahedral and the 13-fold non-icosahedral symmetry of the outer layer led to a reconstruction of the VP6 trimer with clear density for side chains as well as the central Zn and Cl ions and a number of water molecules (*Figure 2*, *Video 1* and *2*). At the estimated resolution of 2.6 Å (*Figure 3A*), atomic model building is possible (*Grigorieff and Harrison, 2011*) and an atomic model obtained by refinement of a previously published structure (*Mathieu et al., 2001*) agrees well with the starting model as indicated by an RMSD of 0.4 Å. Despite good overall agreement, there are a number of significant differences, localized mainly to areas on the periphery of the molecule, which

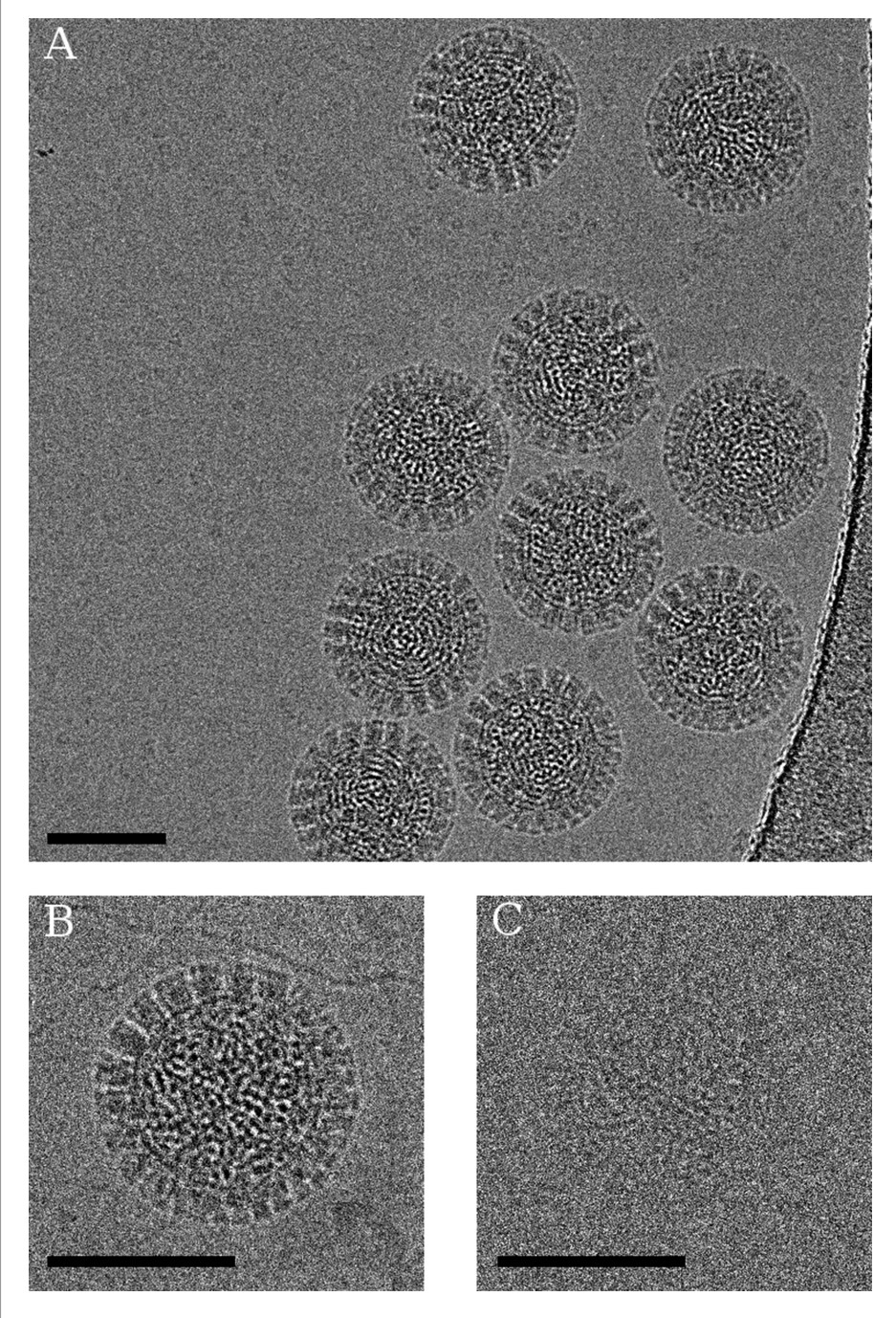

**Figure 1**. (**A**) Example of an aligned movie sum of the rotavirus double-layered particle (DLP) sample imaged with a total exposure of 100 e⁻/Å². (**B**) Particle sum created using all frames and (**C**) the first 3 frames of the movie, demonstrating the kind of data used in the analysis. In all cases, the scale bar represents 500 Å.

may indicate effects of crystal packing and/or the presence of additional protein interactions in the full capsid as opposed to the isolated VP6 subunit.

Calculating the Fourier Shell Correlation (FSC) (*van Heel and Harauz, 1986*) between the VP6 map and the refined atomic model demonstrated a resolution of ~2.7 Å (*Figure 3A*). One possibility for the difference between this and the ~2.6 Å obtained with the half map is the absence of full density for the most radiation-sensitive side chains in the map, while they are present in the model. Assigning

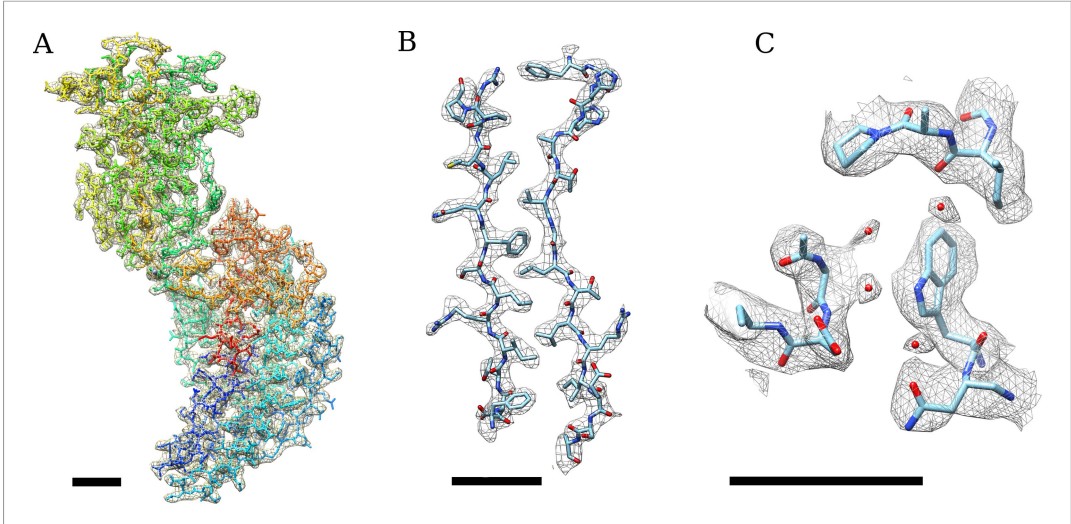

**Figure 2.** (**A**) Density of an isolated VP6 subunit is shown as a mesh along with the docked atomic model. The model is colored using a rainbow spectrum, starting with the N-terminus in blue and ending with the C-terminus in red. (**B**) Zoomed region of the VP6 subunit. (**C**) At higher thresholds, small density features become visible that we interpret as water molecules because their locations are very close to water molecules found in the VP6 crystal structure (**Mathieu et al., 2001**). A B-factor of −175 Å² was applied to the DLP reconstruction before 13-fold non-icosahedral averaging to sharpen the VP6 map. In all cases, the scale bar represents 10 Å.

higher atomic temperature factors to side chains with weak density may accommodate these discrepancies, but we did not explore this in our model refinement.

## Critical exposure curve

A complication of using images for the calculation of critical exposure as opposed to diffraction intensities is the beam-induced movement of the specimen. Previous studies have demonstrated that the magnitude of this movement is not constant across a single exposure, typically being greater at the beginning of the exposure than at the end (**Brilot et al., 2012**; **Campbell et al., 2012**; **Li et al., 2013**). Movement of a particle over the course of the exposure will lead to signal degradation, and if the movement is not constant across the entire exposure, it will lead to an exposure-dependent signal degradation unrelated to radiation damage. Non-constant movement across an exposure will therefore interfere with the critical exposure measurement and must be taken into account. In order to do this, we estimated per-frame shifts for each particle using a new movie alignment program called Unblur (see 'Materials and methods'). During subsequent refinement (see below), we also estimated particle rotations by aligning 3-frame sums (see below). On average, particles rotate by ∼0.9° over the course of a movie, or an average of 0.02° per three frames. A rotation of 0.02° causes a translation of subunits on the surface of DLP of about 0.12 Å, too small to be relevant for our analysis of the fading signal. Movies were recorded such that the specimen was exposed to a total exposure of 100 electrons/Å², split over 130 movie frames (0.77 e⁻/Å² per frame). By the 28th frame (an accumulated exposure of 21 e⁻/Å²), the average particle shift per frame had fallen to 0.2 Å, with 95% of particles shifting by less than 0.54 Å per frame. Between the 28th frame and the 130th frame, the average particle shift per frame remained constant at 0.2 Å, and 95% of the measured particle shift stayed between 0.4 Å and 0.54 Å per frame. Because the movement was almost constant between frames 28–130, we do not expect differences in their signal to be significantly affected by translational movement, and for this reason, only frames 28–130 were used during the analysis.

Movie frames for each particle were aligned individually, but in order to provide more signal, reconstructions were calculated from sums of every 3 movie frames (2.31 e⁻/Å² per sum). Each of these reconstructions of the VP6 trimer was used to calculate an FSC curve (e.g., **Figure 3C**) for the corresponding accumulated exposure. These FSC curves were used to estimate the SNR of a resolution shell ($k$) (**Grigorieff, 2000**):

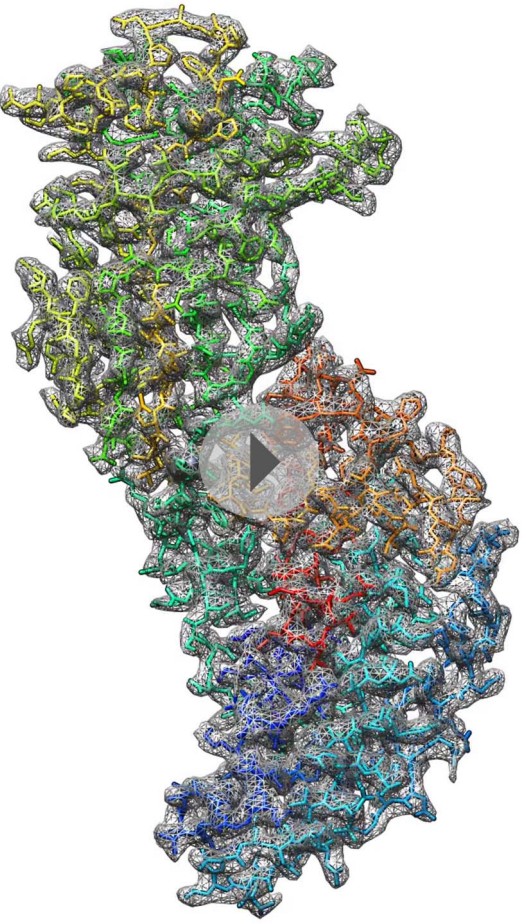

**Video 1.** Density of an isolated VP6 subunit is shown as a mesh along with the docked atomic model (see also *Figure 2A*). The model is colored using a rainbow spectrum, starting with the N-terminus in blue and ending with the C-terminus in red.

$$SNR(k) = \frac{2FSC(k)}{1 - FSC(k)}. \qquad (2)$$

Thus, using the measured FSC curves and *Equation 2*, we were able to plot ln(SNR) for each resolution shell as a function of exposure. For each plot, we performed linear regression using the points from frame 28 up to the point at which ln(SNR) fell below the $3\sigma$ value expected from pure noise. *Figure 4* shows the plots and regressions found for a number of different resolutions.

Good quality regressions were found for data from ~22 Å to ~3.8 Å, and the critical exposure values derived from these regressions are plotted in *Figure 5A*. At resolutions higher than 3.8 Å, there were too few points (estimated SNR values) for reliable fitting. At resolutions below 22 Å, the curves became much noisier resulting in poor quality fits. This may be due to the fact that at resolutions below 22 Å, errors due to inelastic scattering become more significant, and the FSC values at these spatial frequencies are so high (typically greater than 0.999) that small errors lead to large changes in the estimated SNR. Additionally, our VP6 reconstruction has only 4 resolution shells at resolutions lower than 22 Å (nominally at ~27 Å, ~37 Å, 55 Å, and 110 Å). These low-resolution shells thus effectively contain information from a wide range of frequencies, which may also cause errors in the estimation.

We could fit a function to the recorded data, which takes the form

$$y = ax^b + c. \qquad (3)$$

The best fit to the data, with a $R^2$ value of 0.997, was $a = 0.245$, $b = -1.665$, and $c = 2.81$, plotted in *Figure 5A* together with the experimental data. This function can be used to estimate the critical exposure at 300 kV; the critical exposure at 200 kV can be expected to be about 25% lower (*Yalcin et al., 2006*). The optimal exposure, plotted in *Figure 5B*, is ~2.5 times the critical exposure (*Hayward and Glaeser, 1979*).

At higher spatial frequencies (~4 Å), our values agree well with values previously measured on crystalline specimens (*Figure 5B*, *Baker et al., 2010*), but they deviate toward lower spatial frequencies, suggesting that the signal at these frequencies in single-particle images is less sensitive to electron exposure than in images of crystalline specimens. A possible explanation is that when imaging crystalline specimens, there are two modes of damage, a short-range component caused by damage to the molecules themselves and a long-range component caused by loss of crystalline order. We would expect single particles to be affected only by the former and so appear to be less radiation sensitive. Rotavirus DLP may also be affected by lattice distortions because the surface lattice formed by the VP6 trimers may be considered analogous to the lattice of a small crystal. However, the extent to which deformations will attenuate signal will depend on the size of the lattice: small deformations can add up to larger unit cell displacements across larger distances compared to displacements across smaller distances. The crystals used for electron diffraction typically measure 1 μm or more in diameter (*Unwin and Henderson, 1975*), at least 5 times larger than the circumference of DLP. Furthermore,

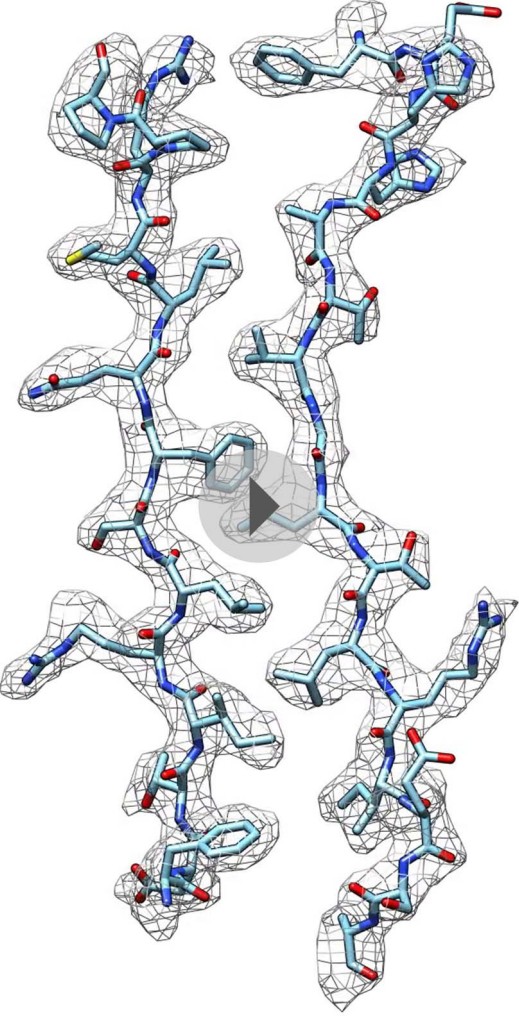

**Video 2.** Zoomed region of the VP6 subunit showing two β-sheet strands (see also *Figure 2B*).

due to their geometry, 2D and thin 3D crystals are prone to in-plane and out-of-plane distortions, both of which can severely attenuate diffraction intensities. It is therefore essential for crystal diffraction studies to use a flat support film, usually amorphous carbon. Studies of beam-induced motion have shown that the observed motion in cryo-EM specimens involves both the ice and the carbon support (*Glaser et al., 2011*; *Brilot et al., 2012*). Some of the beam-induced motion may lead to distortions of the carbon support, causing long-range distortions in the crystals. Beam-induced distortions of the ice layer embedding the DLP particles may also perturb the particle structure. However, the tight packing of the capsids around the RNA genome is likely to yield a particle that is more resistant to distortions than the crystals. We therefore expect that, at least in terms of the structural properties relevant for this study, DLP is much closer to a single particle than a thin crystal. If the icosahedral surface lattice of DLP is also affected by lattice distortions due to radiation damage, we would expect the critical exposure for smaller, less symmetric samples to be higher still. A further difference to the crystal studies is that they used the fastest fading intensities in each resolution bin for their analysis (*Unwin and Henderson, 1975*; *Hayward and Glaeser, 1979*; *Stark et al., 1996*; *Baker et al., 2010*). With a single-particle specimen, we are limited to measuring the average loss of SNR in each resolution bin, which by definition is lower than the fastest fading components. However, the difference between the average and fastest fading components does not appear to be enough to explain the observed differences. Other factors may include the chemical makeup of the specimen, such as the presence of nucleic acids in DLP, as well as buffer and solvent components. However, the effect of these factors on radiation damage is not well understood.

The first 27 frames of the movie were excluded from the analysis so as to prevent beam-induced movement from interfering with the results. Our results are therefore based upon data, which have effectively been pre-exposed by ~20 e−/Å². If radiation damage indeed occurs as a single-exponential process, pre-exposure should have no effect on the result presented here. We cannot exclude the possibility that the damage may occur in two (or more) phases and that we are only observing the final phase, but there is no evidence for a multi-phased process, and previous studies on crystalline specimens also demonstrate a single-exponential decay. Moreover, for lower resolutions at which beam-induced movement is expected to have less of an effect, a plot of ln(SNR) vs accumulated exposure across all frames shows a single-exponential process (see *Figure 4*). We also note that a reconstruction calculated using only frames 25–27 (~19 e−/Å²–~21 e−/Å²) still yields a 3.35 Å resolution reconstruction with clear side chain density (see *Figure 6*), suggesting that the decay we have measured is relevant to the kinds of structural information we are interested in. As expected, the density of side chains fades with increasing exposure and dose deposited on the sample. The density of carboxyl groups (e.g., Asp29, *Figure 6*) fades most rapidly and is already partially gone after an exposure of about 3 e−/Å², while those of aromatic groups (e.g., Tyr24, *Figure 6*) persists even after an exposure of about 35–40 e−/Å². This general pattern agrees with that observed before in a number

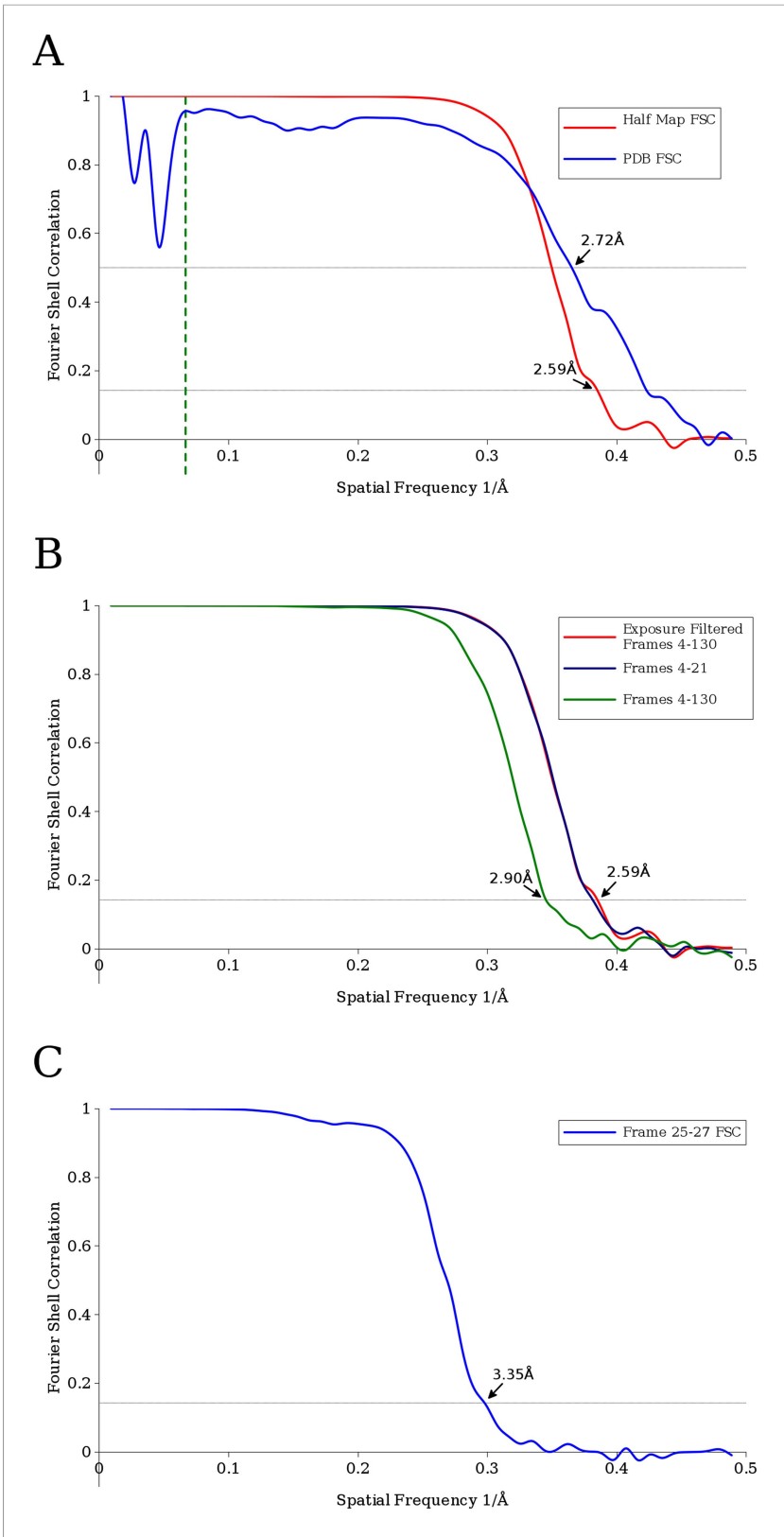

**Figure 3**. (**A**) Fourier Shell Correlation (FSC) curves estimating the resolution of the final VP6 reconstruction calculated using exposure-filtered data. Two FSC curves were obtained, one from maps calculated from two halves of the data set, another using the modeled atomic coordinates. The dashed green line represents 15 Å, the upper

*Figure 3. continued on next page*

*Figure 3. Continued*
resolution limit used during parameter refinement. (**B**) FSC curves between half data set reconstructions of the VP6 subunit when using an exposure-filtered sum of frames 4–130, an unfiltered sum of frames 4–130, and an unfiltered sum of frames 4–21, which were determined to be the best set of unfiltered frames by trial and error. Exposure filtering was applied only to the final reconstruction, and not during refinement (**C**) FSC curve for the reconstruction using only frames 25–27, indicating a resolution of ~3.4 Å after a pre-exposure of ~19 e⁻/Å².

of cryo-EM (*Allegretti et al., 2014*; *Bartesaghi et al., 2014*; *Fromm et al., 2015*) and X-ray crystallography studies (*Fioravanti et al., 2007*). We note also that the density of the α-helix backbone remains visible even after an exposure of ~50 e⁻/Å². This high exposure exceeds the optimal exposure of about 30 e⁻/Å² at 8 Å resolution where α-helical features become clearly visible. Such high tolerance of secondary structural features to radiation damage may help in the alignment of single-particle images recorded using high exposures (see below).

## Exposure filtering

When using a detector, which outputs single images, the optimal exposure curve (*Figure 5B*) can be used to select the optimal exposure based on a targeted resolution, although this exposure will not be optimal for other resolutions. However, as has been suggested before (*Baker et al., 2010*; *Campbell et al., 2012*; *Scheres, 2014*), given a detector capable of recording movies, the optimal exposure curve can be used to filter frames based on their individual exposures. Filtering the frames in this way will result in a sum with an increased SNR relative to the unfiltered sum.

*Equation 1* describes the attenuation of the image SNR caused by radiation damage with increasing exposure. The attenuation of the SNR is the result of fading image Fourier amplitudes (*A*), which will follow the square root of the exponential decay of the SNR:

$$A(k, N) = A(k, 0)e^{-\frac{N}{2N_e(k)}}. \tag{4}$$

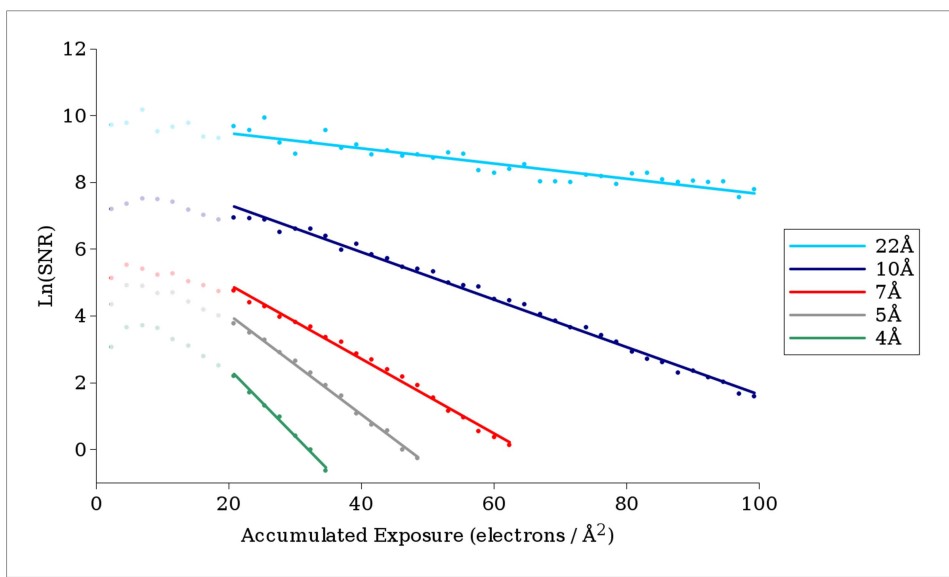

**Figure 4**. Example plots of ln(SNR) vs accumulated exposure with associated linear fits at a number of different resolutions. Data used in this study are shown in darker color, while data for early frames excluded from the analysis due to specimen movement are shown in lighter color. The slopes of the lines become steeper as the resolution increases, corresponding to faster fading of the signal. The linear plots fit the data well, suggesting that in the analyzed regions a single-exponential process is dominant in the decay.

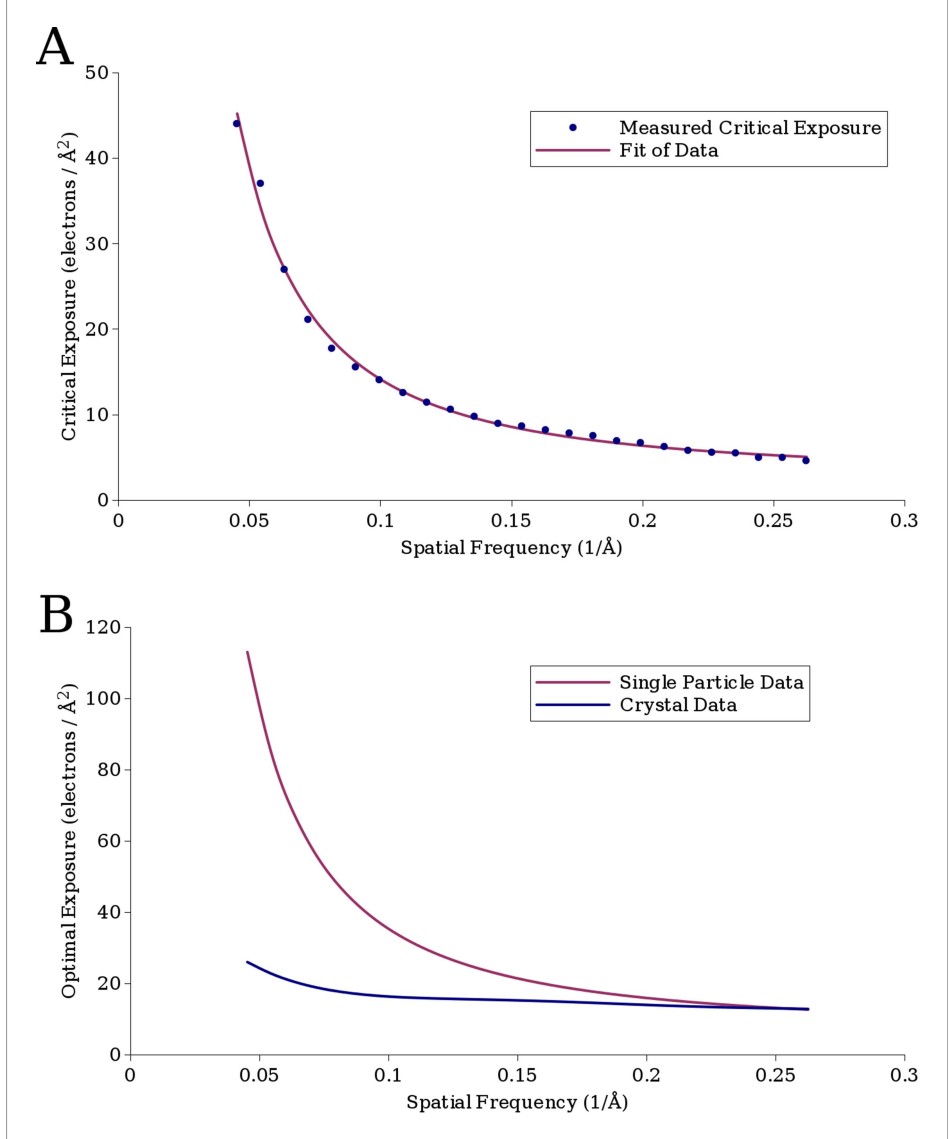

**Figure 5**. (**A**) The measured critical exposure curve plotting the critical exposure at each measured resolution and the fit function for comparison. (**B**) Curve plotting the optimal exposure obtained in this study alongside that obtained in a previous study on crystalline specimens, scaled by a factor of 1.25 to compensate for the fact that the previous study was conducted at 200 kV (**Baker et al., 2010**).

The initial value of $A$ at $N = 0$ is usually not known but is not needed for optimal exposure filtering; only relative differences are significant (see below). Thus, the exposure-dependent amplitude attenuation is given by

$$q(k, N) = e^{-\frac{N}{2N_e(k)}}. \tag{5}$$

Previous work has described the optimal 3D reconstruction from images, which have had their amplitudes attenuated by the CTF (**Sindelar and Grigorieff, 2012**). We can easily extend this formalism to take into account both the CTF and the amplitude attenuation due to radiation damage:

$$F^{3D}(\mathbf{k}) = \frac{\sum_{i=1}^{n} CTF_i^*(\mathbf{k}) q(k, N_i) F_i(\mathbf{k})}{\sum_{i=1}^{n} \left( |CTF_i(\mathbf{k})| q(k, N_i) \right)^2 + 1/SNR(k)}. \tag{6}$$

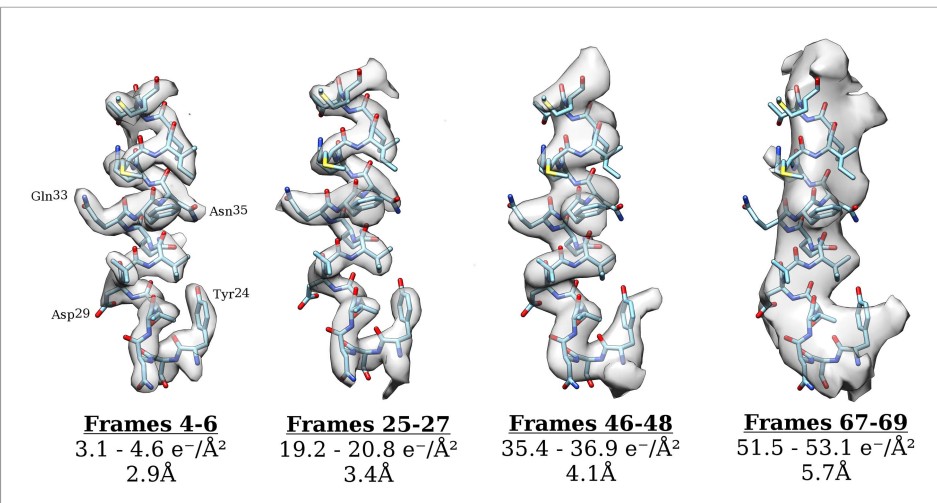

**Figure 6**. Surface rendering of an isolated small helix from different 3-frame reconstructions shown with the docked model. Each reconstruction is shown with its exposure range and resolution as calculated from the FSC using the 0.143 cut-off.

Here, the sum over $i$ includes all frames of all movies contributing to Fourier coordinate $\mathbf{k}$, $F^{3D}$ is the 3D reconstruction Fourier transform, while $F_i$ represents the measured image Fourier terms. $N_i$ is the accumulated exposure of the frame, and SNR($k$) is the average unattenuated SNR in the area of the particle within a single frame (the 'particle spectral SNR', or PSSNR in [*Sindelar and Grigorieff, 2012*]).

While *Equation 6* describes an optimal correction when 3D reconstructions are calculated from individual movie frames, it is often more practical to calculate a filtered sum of the frames for each aligned movie or individual particle. This sum can then be used for further processing without further consideration of the movie frames. In this case, for simplicity, we assume that the defocus does not change and thus disregard the CTF term, leaving it to be taken into account in later processing steps. The filtered 2D image $F^W$ is then given by

$$F^W(\mathbf{k}) = \frac{\sum_{i=1}^{n} q(k, N_i) F_i(\mathbf{k})}{\sum_{i=1}^{n} q^2(k, N_i) + 1/\text{SNR}(k)}. \tag{7}$$

SNR($k$) is not known, but we can assume it to be very small for a single frame of a movie due to the small exposure. Its reciprocal value is therefore large compared to the first term in the denominator, which can be neglected, and thus

$$F^W(\mathbf{k}) \approx \text{SNR}(k) \sum_{i=1}^{n} q(k, N_i) F_i(\mathbf{k}). \tag{8}$$

SNR($k$) is an average across a resolution shell, and thus, the term serves to weight resolution shells with respect to each other to maximize the SNR of the final image. This will generally result in a low-pass filtering of the sums and may affect later processing steps. Furthermore, an estimate of the SNR is usually not available until later stages of processing. Thus, to minimize the alteration of features in the original images by the exposure filter, we formulate the filter as a weighted sum that maximizes the SNR in each resolution shell:

$$\bar{F}^W(\mathbf{k}) = \frac{\sum_{i=1}^{n} q(k, N_i) F_i(\mathbf{k})}{\sqrt{\sum_{i=1}^{n} q(k, N_i)^2}}. \tag{9}$$

Exposure filtering using the above scheme, combined with the measured critical exposure curve, has been integrated into the new Unblur program (see 'Materials and methods') allowing the production of aligned and exposure-filtered movies. Increasing the SNR of the movie sum should increase the accuracy of particle alignments leading to improved reconstructions, particularly in the most difficult cases such as very small proteins. Furthermore, correct filtering of movie frames allows

the use of all later frames in the reconstruction with no need to test the inclusion of different numbers of frames in order to find an optimum.

In order to assess the effectiveness of applying the exposure filter, we first used trial and error to manually determine which range of frames, when summed, yielded the best VP6 reconstruction. The best reconstruction was found when using frames 4–21. Separately, we calculated a reconstruction using frames 4–130 (i.e., to the final frame of the movie), weighted with the exposure filter. The two reconstructions are nearly identical as judged by the FSC and are both of better quality than a reconstruction using the unfiltered sum of frames 4–130, suggesting that the exposure filter is weighting correctly (see *Figure 3B*).

While the reconstructions calculated using exposure filtering and by selecting frames 4–21 have the same resolution, the exposure-filtered reconstruction visually appears slightly less sharp. This higher B-factor is to be expected due to the inclusion of comparatively stronger lower resolution signal in the exposure-filtered sums when compared to the simple sum of frames 4–21. After scaling the amplitudes to be the same for both reconstructions using diffmap (http://grigorieflab.janelia.org/diffmap), the two reconstructions are indistinguishable. In the case of DLPs, which exhibit extremely low-alignment error using traditional processing (*Henderson et al., 2011*), increasing the SNR at low and intermediate resolutions would not be expected to lead to improved alignment. However, additional resolution gains can be expected for smaller particles with a lower molecular mass, when using exposure-filtered particle sums (see below).

## Application of the exposure filter to 20S proteasome

Using recently published data to calculate a 2.8 Å reconstruction of the 20S proteasome, a 700 kDa complex with D7 symmetry (*Campbell et al., 2015*), we applied the exposure filter described above to test if it also optimizes the reconstruction of a smaller, less symmetrical particle. Movies of 20S proteasome consisted of 38 frames collected on a Gatan K2 detector using a total exposure of 53 e$^-$/Å$^2$ (~1.4 e$^-$/Å$^2$ per frame). Using the frame alignment obtained by Campbell et al., we calculated four different frame sums: using the exposure filter, unfiltered sums of the first 9 frames (corresponding to an effective exposure of 12.6 e$^-$/Å$^2$), the first 14 frames (corresponding to an effective exposure of 19.6 e$^-$/Å$^2$), and all frames. An exposure of 12.6 e$^-$/Å$^2$ approximates the optimal exposure at ~3 Å resolution measured using rotavirus DLP, while an exposure of ~20 e$^-$/Å$^2$ is close to exposures used in previous high-resolution cryo-EM studies (*Zhang et al., 2008*; *Wolf et al., 2010*) that were published prior to the use of direct detectors and movies. The same particles selected and used by Campbell et al. (49,954 particles) were extracted from these four sets of micrographs, aligned against an initial model obtained from class averages (see 'Materials and methods') and refined using Frealign v9 (*Lyumkis et al., 2013*).

A comparison between the previously published structure and our structure (*Figure 7A*) suggests that the exposure filter described here and the B-factor weighting implemented in Relion (*Scheres, 2014*; see below) and used by Campbell et al. perform equally well. *Figure 7B* shows FSC curves for all four test cases, indicating a resolution of 2.8 Å for all reconstructions except the one calculated from unfiltered frame sums that included all 38 frames. The lower resolution of this last reconstruction (about 3 Å) is expected due to the effective total exposure of 53 e$^-$/Å$^2$, which exceeds the optimal exposure at 3 Å by a factor of 4. The optimal exposure is also exceeded in the case of 19.6 e$^-$/Å$^2$, albeit only by a factor of 1.5, which does not appear to affect the resolution of the reconstruction significantly. To better follow the effectiveness of the exposure filter, we also plotted the average spectral SNR of the filtered particle images (PSSNR, see above). *Figure 7C* shows that the SNR of the exposure-filtered sums of all frames equals or exceeds that of the other frame sums at all resolutions. Without exposure filtering, summing all 38 frames produces SNR values that are similar to the filtered sums at low resolution but are significantly lower as the resolution increases. At 3 Å resolution, the average SNR in the unfiltered sums is only about 30% of that of the filtered sums. The PSSNR curves estimated for the unfiltered sums, calculated from the first 9 and 14 movie frames, start out lower at low resolution than the other two curves as expected since they are missing the additional exposure to boosts the low-resolution signal. At 3 Å resolution, both curves converge with the curve calculated form the filtered sums using all 38 frames. Based on the effective radiation damage reflected in the data, we would expect the PSSNR at high resolution to be highest for the exposure-filtered sums, followed by the unfiltered sums corresponding to 12.6 e$^-$/Å$^2$, 19.6 e$^-$/Å$^2$, and 53 e$^-$/Å$^2$. However, in

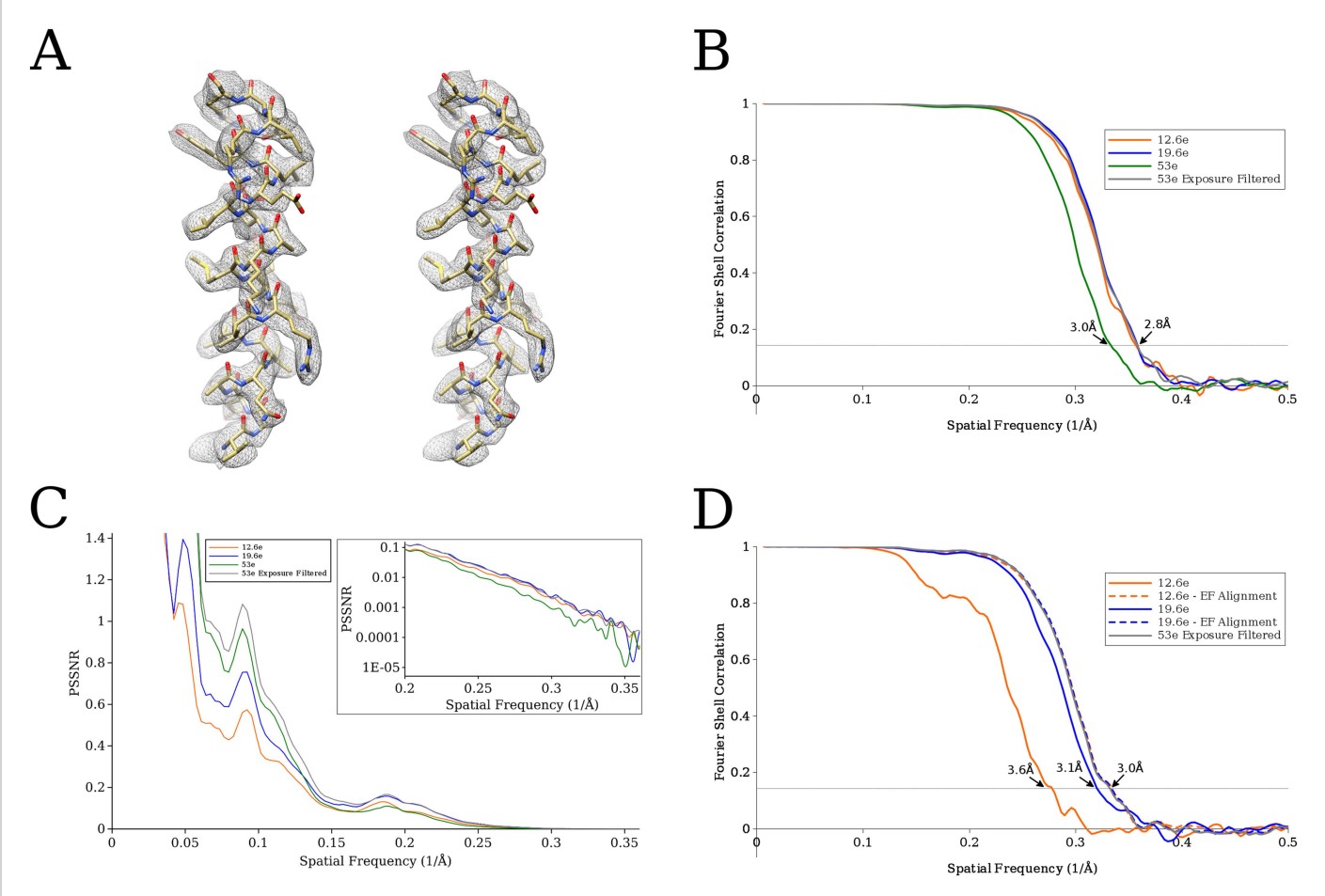

**Figure 7**. (**A**) Comparison of an isolated helix from the previously published reconstruction (*Campbell, 2015*) on the left, and the reconstruction using exposure-filtered data on the right. The two maps appear almost identical after scaling the amplitudes using diffmap (http://grigoriefflab.janelia.org/diffmap), suggesting that in this case exposure filtering performs as well as the weighting based on B-factors implemented in Relion (*Scheres, 2014*). (**B**) Plot of FSC curves for the various proteasome reconstructions. The exposure-filtered reconstruction has a resolution of ~2.8 Å, matching the resolution previously obtained. (**C**) Plot of the average particle signal-to-noise ratio (SNR) as a function of resolution. The exposure-filtered particles have equal or higher SNR than the other data sets at all resolutions. (**D**) Plot of FSC curves from the signal-limited data set. In this case, the exposure-filtered reconstruction is of better quality than those derived from the other data sets. Recalculating the non-filtered reconstructions with particle alignment parameters obtained for the exposure-filtered data set increases the resolution to that of the filtered data set (curves labeled 'EF Alignment'), demonstrating that the loss in resolution was due to particle misalignments.

practice, the estimates are subject to some error due to the weak signal at high resolution and the fact that all data sets were independently refined from low resolution (see 'Materials and methods'). Therefore, only the data set with the most significant signal attenuation (sums of 38 unfiltered frames) deviates significantly from the rest, and the remaining three curves indicate very similar SNR values.

## Improved alignment when signal is limiting

The higher SNR present in exposure-filtered frame sums, compared with unfiltered sums containing a lower total exposure, is expected to help in the alignment of smaller particles where the alignment accuracy is significantly limited by the signal generated by the particle. The benefit of a higher total exposure is not apparent in the alignment of rotavirus DLP because the alignment errors of all tested frame sums is extremely small. Even the resolution of the reconstruction of the 20S proteasome with a total mass of 700 kDa does not appear to be limited by alignment errors at exposures of 12.6 e$^-$/Å$^2$ and higher (see previous section). To demonstrate improved alignment at higher total exposures

when the signal is weak, we generated additional data sets by adding images of areas of empty ice to the proteasome particle images. The ice images were taken from the same micrographs as the particles they were added to, and they contained the same number of frames (total exposure) and filtering. The ice images add approximately the same amount of noise and background that is also present in the particle images, thus, reducing the effective SNR in the resulting images approximately by 50%. At low resolution (special frequencies larger than 1/10 Å$^{-1}$), the new data sets should therefore emulate data that would have been obtained by a particle of half the mass, that is, 350 kDa, while at higher resolution (special frequencies smaller than 1/10 Å$^{-1}$) the data should approximate the signal expected from a 175 kDa particle, that is, with a quarter of the mass (*Rosenthal and Henderson, 2003*).

*Figure 7D* shows the results of the processing of three data sets with added ice background, derived from exposure-filtered sums of 38 frames, as well as unfiltered sums of the first 9 and 14 frames (total exposures of 12.6 e$^-$/Å$^2$, 19.6 e$^-$/Å$^2$, respectively). Unlike the reconstruction calculated from the exposure-filtered original data (see above), the reconstruction calculated form the exposure-filtered images with added noise displays a significantly improved FSC curve compared with the other two noise-added data sets. To test if the improved resolution with exposure filtering is due to the improved alignment of the images, we also calculated reconstructions from the noise-added data sets with total exposures of 12.6 e$^-$/Å$^2$, 19.6 e$^-$/Å$^2$ using the alignment parameters obtained from the exposure-filtered data. As shown in *Figure 7D*, this parameter replacement increases the resolution for the two unfiltered reconstructions to that of the filtered reconstruction, similar to the results obtained for the original data sets in *Figure 7B*. The superior resolution of the exposure-filtered reconstruction is therefore due to more accurate particle alignments. High-quality data sets of smaller particles (about 300 kDa or less) collected using 50 to 100 e$^-$/Å$^2$ exposures are currently not available to demonstrate the improved particle alignment more directly with experimental data. However, the alignment accuracy of these particles is expected to be limited by weaker contrast (*Henderson, 1995*), and we expect improvements similar to those seen in our simulation when using exposures that significantly exceed the optimal exposure and applying the exposure filter described here.

## Discussion

The optimal exposure curve we determined should correct well for radiation damage to the specimen, which has a substantial effect on the relative signal content of individual movie frames, in particular the later frames. As discussed earlier, the other major source of frame-dependent signal change is motion of the specimen, which tends to degrade the signal of earlier frames. For optimal filtering of the data, movement should also be included in the filtering, as should any other effects that change the relative signal in different frames. Currently, the exposure filter does not take movement into account and so will not filter the initial frames optimally. We plan to incorporate movement in a future version of the algorithm.

An alternative method for frame filtering has recently been described (*Scheres, 2014*). This methodology uses the data set itself to estimate per frame weights based on fitting relative B-factors to reconstructions from individual frames. Although this is an elegant solution, which in the best case handles weighting of both radiation damage and movement, filtering with the exposure curve described in this paper should offer a number of advantages, at least in terms of filtering radiation damage. First, our filter can be applied at the beginning of processing without assessing the data, and thus, can help in the initial steps of picking particles and finding initial alignment and orientation parameters. Indeed, one could use the two methods in combination, by using the exposure filter described here for the initial refinement, then when a good quality reconstruction has been obtained, using frame filtering based directly on the data. Second, for some data, the estimate for the B-factor to be applied to each frame may be quite noisy and may lead to error, a problem, which is likely to be exacerbated with smaller data sets. Furthermore, accurate estimation of the B-factor relies on high-frequency signal and will thus fail on intermediate- and low-resolution structures. Third, even in situations in which a B-factor can be estimated, it is unclear how well a B-factor estimated using a specific resolution range will describe signal degradation due to radiation damage outside that range. In contrast, our calculated optimal exposure curve describes experimentally determined values at a wide range of resolutions. The tests carried out with the previously published 20S proteasome data (see above) show that the filtering is indeed close to optimal, not only for rotavirus DLP but also for smaller and less symmetrical particles, and that its performance matches that of B-factor

weighting. Finally, exposure filtering can also be applied when B-factors are difficult to determine, such as in electron tomography. Using the filter described here, images collected at different specimen tilts can be filtered to optimize the final SNR and resolution in a tomogram. While a similar filtering could also be achieved using B-factors, the relatively low resolution of a tomogram and the presence of structural heterogeneity would likely prevent the determination of appropriate B-factors.

The results we present here indicate that the electron exposures commonly used in cryo-EM experiments ($\sim$10–20 e$^-$/Å$^2$) should be increased to obtain optimal image contrast. By using a considerably larger total exposure, the SNR of the images at intermediate and low resolutions will be increased, leading to greater accuracy in particle alignment and orientation determination in cases where the accuracy is limited by the signal, which will in turn lead to better reconstructions. This should especially be true for smaller particles where alignment errors can severely limit the attainable resolution (*Henderson et al., 2011*).

# Materials and methods

## Specimen preparation

Rotavirus DLPs were prepared as previously described (*Street et al., 1982*). Three microliters of sample with a concentration of 2.5–4 mg/ml was applied to C-flat 1.2/1.3 Cu 400 mesh grids (Protochips, Raleigh, NC), which had previously been subjected to glow discharge for 45 s at 20 mA, and plunge-frozen using an FEI Vitrobot Mark 2 (FEI Company, Hillsboro, OR) with a 4 or 6 s blot time and at relative humidity between 65 and 80%.

## Electron microscopy

The data were collected on an FEI Krios microscope (FEI Company, Hillsboro, OR) operating at 300 kV. Movies were collected on a Gatan K2 Summit direct electron detector (Gatan, Inc., Pleasanton, CA) in super resolution mode with a calibrated pixel size of 0.512 Å per super resolution pixel. The pixel size was calibrated by maximizing the cross correlation between the whole DLP reconstruction and a 3.8 Å crystal structure of the entire DLP (*McClain et al., 2010*). Each exposure was 13 s long and recorded as a movie of 130 frames. The exposure per frame as reported by Digital Micrograph (Gatan, Inc., Pleasanton, CA) was 0.769 e$^-$/Å$^2$, which corresponds to an exposure of 8 electrons/pixel/s on the camera. Movies were collected at a range of underfocus between $\sim$0.4 µm and $\sim$2.0 µm. Throughout the data collection, the exposure per movie was checked regularly to make sure it hadn't deviated from a total exposure of 100 e$^-$/Å$^2$.

## Image processing

Super-resolution movie frames were initially corrected for a magnification distortion present on our FEI Titan Krios microscope by real space stretching using bilinear interpolation. The frames were then downsampled by a factor of 2 using Fourier cropping to a pixel size of 1.023 Å. The downsampled frames were motion-corrected using newly written software called Unblur (*Supplementary file 1*). Unblur is stand-alone software available for download from the Grigorieff lab web page (http://grigoriefflab. janelia.org/unblur) and is based on iterative alignments of each raw frame to the current best total sum of all other frames, leaving the frame which is currently being aligned out of the total sum to avoid the frame 'finding itself' in the sum. After each iteration, a spline is fit to both the X and Y shifts to reduce sensitivity to noise. Frame sums can be filtered according to the described exposure filter, or the filtering step can be skipped. Frame sums of already aligned frames can be recalculated using the program Summovie (*Supplementary file 2*), which is also available for download from the Grigorieff lab web page (http://grigoriefflab.janelia.org/unblur). The filter constants (*Equation 3*) are not user accessible but can easily be changed in the source code if needed.

4178 DLP particles were picked manually from the aligned movie sums, and extracted into 1024 × 1024 boxes. Filtered amplitude spectra were calculated for each particle and were used to estimate the defocus and astigmatism values on a per-particle basis using the FindCTF program of the TIGRIS package (http://tigris.sourceforge.net/). TIGRIS consists of a set of programs for single-particle image analysis, including algorithms for CTF determination and correction, image alignment and classification, 3D reconstruction and a fully-featured display. FindCTF attempts to determine the optimal defocus parameters by maximizing the correlation between an amplitude spectra and

a theoretical CTF, first performing a brute-force parameter search followed by downhill simplex optimization. Particles were then individually motion corrected using the Unblur algorithm on particles boxed from individual frames. An initial model was calculated from a single DLP image using Angular-Reconstitution in the IMAGIC package (*van Heel et al., 1996*). Initial parameters were obtained on images resampled to 8 Å per pixel via Fourier cropping, which were aligned to the initial model with the brute force alignment program of the TIGRIS package. This alignment finds the highest cross-correlation peak across a specified set of in-plane rotations (in this case every 1°) of reference projections (in this case projections of the model sampled at every 5°). These parameters were further refined using Frealign v9 (*Lyumkis et al., 2013*) using data sampled at 1.023 Å per pixel, and using information between 200 Å and 15 Å.

In order to increase the signal present in the raw frames, 3-frame sums of the motion corrected particles were calculated, and these sub-sums were individually refined using Frealign again using data between 200 Å and 15 Å. Individual reconstructions were calculated from each set of 3 frame sums, providing 43 individual DLP reconstructions at increasing exposure. Frealign outputs two half map reconstructions used for calculating an FSC curve. For the two half maps of each of the 43 reconstructions, additional 13-fold averaging of the VP6 trimer was performed using the AVE program (*Kleywegt and Read, 1997*) and matrices derived from the asymmetric unit of the previously published 3.8 Å crystal structure of the DLP (*McClain et al., 2010*). The resulting VP6 half maps were masked with a soft-shaped mask and used to calculate FSC curves, resulting in 43 curves (e.g., *Figure 3C*) each corresponding to a different exposure, which were used for the critical exposure estimation.

In order to obtain the highest resolution reconstruction, subsets of frames were manually selected and a trial and error approach resulted in the determination that refinement using the sum of frames 4–21 and again the resolution range of 200 Å to 15 Å resulted in the highest resolution reconstruction as determined by the FSC between two half maps masked with a soft-shaped mask. The resulting resolution as determined by the 0.143 cut-off (*Rosenthal and Henderson, 2003*) was 2.59 Å (*Figure 2A*). Maps were rendered using UCSF Chimera (*Pettersen et al., 2004*). To render specific sections of a map, a zone including specified amino acid residues within the fitted atomic model was defined, and the only densities within a radius of 3 Å of the zone were displayed (command 'zone' in Chimera). Furthermore, disconnected density was removed from the display (command 'hide dust' in Chimera) in maps shown in *Figures 2A, 6* (except for the first panel), and *Figure 7A*.

An initial fit of a previously determined crystal structure (*Mathieu et al., 2001*) fit well for the majority of the structure; however, it was clear that some regions needed adjustment, and therefore, real-space refinement was carried out in Coot (*Emsley et al., 2010*) exploiting all restraints. As a further validation, the resulting model was converted into a density map using the UCSF Chimera package (*Pettersen et al., 2004*) specifying 2.5 Å resolution, and an FSC curve between this density map and the optimum reconstruction was calculated (*Figure 2A*). The resolution as determined by the 0.5 cut-off (*Rosenthal and Henderson, 2003*) was 2.72 Å.

## Analysis of the 20S proteasome data set

Aligned micrographs were downloaded from the EMPIAR database (EMPIAR-10023), particle locations, and estimated defocus parameters corresponding to the best published reconstruction (*Campbell et al., 2015*) were kindly provided by the authors. Particle sums corresponding to 12.6 e$^-$/Å$^2$, 19.6 e$^-$/Å$^2$, and the total sum of 53 e$^-$/Å$^2$ were calculated, along with particle sums for the whole 53 e$^-$/Å$^2$ weighted with the exposure filter. Signal-limited particle sums were created as follows. First an area devoid of particles was selected from each movie sum. These were then extracted from movie sums corresponding to 12.6 e$^-$/Å$^2$, 19.6 e$^-$/Å$^2$, and 53 e$^-$/Å$^2$ weighted with the exposure filter. Particle sums corresponding to each of these total exposures were then added to the empty area corresponding to the movie sum they were extracted from.

Class averages were generated using MSA within IMAGIC and the exposure-filtered particles. ~10 of these class averages were manually selected and given as input to e2initialmodel.py from the EMAN2 package (*Tang et al., 2007*) to generate an initial model. This initial model was used as a starting model for all subsequent refinements. Processing for all data sets followed the following procedure. Particles sampled to a 5.3 Å pixel size were aligned to the initial model using the TIGRIS brute-force alignment program. These initial parameters were then refined using Frealign, starting with a resolution cut-off of 10 Å and gradually increasing to a resolution cut-off of 5 Å for the final

rounds. For all reconstructions, FSCs were calculated between the Frealign output half maps after first masking with a soft-shaped mask.

## Acknowledgements

The authors thank Stephen Harrison and members of his laboratory for rotavirus DLPs and for comments on the manuscript, Axel Brilot for providing access to frozen DLP grids, Peter Rosenthal for comments on the manuscript, and Alexis Rohou for many valuable discussions and comments on the manuscript. The 20S proteasome data were collected at the National Resource for Automated Molecular Microscopy, which is supported by a grant GM103310 from the National Institute of General Medical Sciences. We thank Melody Campbell for supplying the particle coordinates used in our proteasome tests.

## Additional information

### Funding

| Funder | Grant reference | Author |
|---|---|---|
| Howard Hughes Medical Institute (HHMI) | | Timothy Grant, Nikolaus Grigorieff |
| National Institute of General Medical Sciences (NIGMS) | GM103310 | Nikolaus Grigorieff |

The funders had no role in study design, data collection and interpretation, or the decision to submit the work for publication.

### Author contributions

TG, Conception and design, Acquisition of data, Analysis and interpretation of data, Drafting or revising the article; NG, Conception and design, Analysis and interpretation of data, Drafting or revising the article

## Additional files

### Supplementary files

• Supplementary file 1. Archive containing the program Unblur. It contains the program precompiled for use on a Linux 64-bit system and source code. This archive and future updates are also available for download from the Grigorieff lab web page (http://grigoriefflab.janelia.org/unblur).

• Supplementary file 2. Archive containing the program Summovie. It contains the program precompiled for use on a Linux 64-bit system and source code. This archive and future updates are also available for download from the Grigorieff lab web page (http://grigoriefflab.janelia.org/unblur).

### Major datasets

The following datasets were generated:

| Author(s) | Year | Dataset title | Dataset ID and/or URL | Database, license, and accessibility information |
|---|---|---|---|---|
| Grant T, Grigorieff N | 2015 | Single particle cryo-EM structure of rotavirus VP6 at 2.6 Angstrom resolution | http://www.rcsb.org/pdb/explore/explore.do?structureId=3J9S | Publicly available at RCSB Protein Data Bank (Accession No. 3J9S). |
| Grant T, Grigorieff N | 2015 | Single particle cryo-EM structure of rotavirus VP6 at 2.6 Angstroms resolution | http://www.ebi.ac.uk/pdbe/entry/emdb/EMD-6272 | Publicly available at Electron Microscopy Data Bank (Accession No. 6272). |

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
