## [Decision Letter]

Thank you for sending your work entitled “Measuring the optimal exposure for single particle cryo-EM using a 2.6 Å reconstruction of rotavirus VP6” for consideration at *eLife*. Your Tools and Resources article has been favorably evaluated by John Kuriyan (Senior editor), a Reviewing editor, and three reviewers.

The following individuals responsible for the peer review of your submission have agreed to reveal their identity: Wes Sundquist (Reviewing editor); Janet Vonck (peer reviewer) and John Rubinstein (peer reviewer). A further reviewer remains anonymous.

The Reviewing editor and the reviewers discussed their comments before we reached this decision, and the Reviewing editor has assembled the following comments to help you prepare a revised submission.

Grant and Grigorieff describe an exciting next step in the recent spectacular series of cryo-EM maps from direct electron detector data, with a reconstruction of the rotavirus VP6 protein at a resolution of 2.6 Å from 4000 virus particles, using both the icosahedral symmetry and the 13-fold non-icosahedral symmetry (more than three million asymmetric units). The large size of the particles made it possible to analyse individual movie frames, and a 130-frame, 100 e-/Å2 data set was used to assess the effects of radiation damage. The authors conclude that optimal doses for EM maps at low and intermediate resolutions are much higher than the 10-20 e-/Å2 determined previously using crystalline samples, for example with a pre-exposure of 50 e-/Å2 a map at better than 6 Å resolution could be obtained. The authors speculate that the loss of long-range order in the crystalline samples may have played a role. Making use of the fact that the rotavirus particles can be aligned well from very short exposures (they use the average of 3 frames) they devise a filtering scheme for the data based on the resolution-dependent SNR fall-off as a function of exposure, and show that using this scheme a 2.6 Å reconstruction can be done from the full 130-frame dataset that is indistinguishable from the best reconstruction using unweighted frames 4-21 only. The dose filtering scheme has been integrated in a new program for frame alignment.

The presented data are beautiful, the paper is clear and well-written, the work provides welcome additions to the growing toolbox available to the still young field of cryo-EM by direct electron detectors. Overall, this is an excellent paper of high value and high interest to the scientific community.

Significant issue that the authors must address

1) The dose optimized weighting scheme applied to the entire movies (frames 4-130) produces exactly the same resolution as when using a much smaller subset of the data (frames 4-21) determined by manually searching for the ideal dose fractionation set of frames. While it is convenient to be able to avoid the tedious manual search, it is a harder sell to justify acquiring 5x the number of frames (all needing storage and processing etc.) to get to the same resolution. The authors are no doubt aware of this—but ought to address it directly in the paper—and thus base the justification for the high dose acquisition and weighting scheme on the proposition that this will have a much more important impact on smaller particles. This could then justify the pain of acquiring many more frames but this remains to be proved. Can the authors use one of the publicly available datasets to prove this proposition (e.g. the raw movie frames and other metadata of the proteasome dataset, which goes to 2.8A resolution, are available on EMPIAR)? If the authors decide this is too much work then the emphasis on this point should be reduced, and the authors should modify the statement that the major impact of the paper is enabling SPEM on smaller particles (as this is not yet shown). Instead, the authors could choose to place more emphasis on the extraordinarily high resolution of the reconstruction that has been achieved here. This might involve including some discussion on how the data was obtained and what methods were used to reach the exceptionally higher resolution.

Other issues that the authors should address:

1) The authors base their analysis of radiation damage on SNR values calculated from the FSC. Is this really equivalent to the critical exposure (*N*_*e*_) values measured by following the fading of the calculated diffraction spots from 2D crystals? (The answer may indeed be yes, but a little justification is needed).

2) The authors convert from critical exposures to optimal exposures using the Nopt=∼2.5Ne relationship from [18]. This ratio is based on a rather complex derivation and several assumptions. The authors should justify its use here, showing or explaining how it still applies to the Ne values measured from SNRs and the FSC.

3) The dose filtering was tested on the same data that was used to derive the critical dose curve. However, the authors speculate that the rotavirus with 780 copies of VP6 may to some extent be affected by some loss of long-range order as a function of dose like the 2D crystal samples. If this is the case, their dose curve would overestimate the radiation sensitivity of smaller complexes. This issue should be discussed.

4) The details of the new program Unblur are vague. Is it available? Is it stand-alone or part of a package? Is the dose filtering optional and (in light of the previous comment) are the values hard-wired? It is important to add this information so the new program can be tested by the community.

---

## [Author Response]

*The presented data are beautiful, the paper is clear and well-written, the work provides welcome additions to the growing toolbox available to the still young field of cryo-EM by direct electron detectors. Overall, this is an excellent paper of high value and high interest to the scientific community*.

Significant issue that the authors must address

*1) The dose optimized weighting scheme applied to the entire movies (frames 4-130) produces exactly the same resolution as when using a much smaller subset of the data (frames 4-21) determined by manually searching for the ideal dose fractionation set of frames. While it is convenient to be able to avoid the tedious manual search, it is a harder sell to justify acquiring 5x the number of frames (all needing storage and processing etc.) to get to the same resolution. The authors are no doubt aware of this—but ought to address it directly in the paper—and thus base the justification for the high dose acquisition and weighting scheme on the proposition that this will have a much more important impact on smaller particles. This could then justify the pain of acquiring many more frames but this remains to be proved. Can the authors use one of the publicly available datasets to prove this proposition (e.g. the raw movie frames and other metadata of the proteasome dataset, which goes to 2.8A resolution, are available on EMPIAR)? If the authors decide this is too much work then the emphasis on this point should be reduced, and the authors should modify the statement that the major impact of the paper is enabling SPEM on smaller particles (as this is not yet shown). Instead, the authors could choose to place more emphasis on the extraordinarily high resolution of the reconstruction that has been achieved here. This might involve including some discussion on how the data was obtained and what methods were used to reach the exceptionally higher resolution*.

We toned down our statement that the high-exposure technique described here will benefit smaller particles. In addition, we have applied the exposure filter to the recently published proteasome data as suggested by the reviewers, and we also tested its performance in a situation where the alignment accuracy is significantly limited by the strength of the signal present in the images. For the latter, we added to the proteasome images background noise that was extracted from the proteasome micrographs. The resulting images should therefore resemble particle images with about half of the original SNR, similar to what would be expected from a smaller particle of between 175 – 350 kDa mass. In this test, the reconstruction calculated from the exposure-filtered images (with appropriate ice background added) exhibit a significantly higher resolution than reconstructions of non-filtered images with added ice background. We show that the resolution difference is due to different accuracies in the alignment. We describe the results in two new sections: “Application of the exposure filter to 20S proteasome” and “Improved alignment when signal is limiting”.

*Other issues that the authors should address*:

*1) The authors base their analysis of radiation damage on SNR values calculated from the FSC. Is this really equivalent to the critical exposure (*N_e_*) values measured by following the fading of the calculated diffraction spots from 2D crystals? (The answer may indeed be yes, but a little justification is needed)*.

The critical exposure values describe the exponential decay of the diffraction intensities obtained from a crystal. When non-crystalline sample (i.e. single particles) are used, diffraction intensities are replaced by intensities calculated from the Fourier transforms of the images. As explained in [18], the fading intensities reduce the SNR present in the affected diffraction spots. The same will be true for the intensities derived from the image Fourier transforms. The SNR is defined as the ratio of the intensity of the spot/squared Fourier amplitude (signal) and variance (noise) of the intensity due to the stochastic events of incident electrons forming the diffraction pattern/image. The magnitude of the intensity variance is determined by the number of electrons contributing to an exposure. When the exposure is evenly divided into frames of a movie as described here, each frame receives the same number of electrons and therefore, the variance of the intensities will remain constant from frame to frame. The fading of intensities is therefore directly proportional to the reduction of SNR. As explained in the manuscript, instead of following the intensities in Fourier transforms, the SNR can also be calculated from the FSC, making it equivalent to the measurement of fading diffraction intensities. We have added additional justification to the manuscript (Introduction section).

*2) The authors convert from critical exposures to optimal exposures using the Nopt=∼2.5Ne relationship from*
[18]*. This ratio is based on a rather complex derivation and several assumptions. The authors should justify its use here, showing or explaining how it still applies to the Ne values measured from SNRs and the FSC*.

Based on the equivalence of following the fading of diffraction intensities and evaluating FSC curves at different total exposures, the original calculation of Nopt = ∼2.5 Ne from [18] also applies. It is simply given by maximizing

SNR ∼ (1−e−N2Ne)2N which occurs at Nopt=2.51284 Ne . We now point this out in the text in the Introduction section.

*3) The dose filtering was tested on the same data that was used to derive the critical dose curve. However, the authors speculate that the rotavirus with 780 copies of VP6 may to some extent be affected by some loss of long-range order as a function of dose like the 2D crystal samples. If this is the case, their dose curve would overestimate the radiation sensitivity of smaller complexes. This issue should be discussed*.

We already state in the subsection “Critical exposure curve” that if the radiation damage causes disorder in the icosahedral lattice in addition to the damage of the protein, our critical dose measurements will underestimate the critical dose for a smaller, less symmetrical particles. The main parameter to consider here is the size of the object. Larger objects will be more strongly affected by small distortions as these can add up across large distances to large shifts of subunits. We have now added a discussion of differences between rotavirus DLP and crystals, including how likely either sample is affected by long-range disorder. Furthermore, we have added in the Discussion that there may be other factors affecting the critical dose measurements, for example the presence of nucleic acids (RNA in the case of rotavirus DLP), buffer conditions and particle density.

*4) The details of the new program Unblur are vague. Is it available? Is it stand-alone or part of a package? Is the dose filtering optional and (in light of the previous comment) are the values hard-wired? It is important to add this information so the new program can be tested by the community*.

The new program Unblur is already available upon request and will be posted on the Grigorieff lab web page upon publication of this work. It is a stand-along program that uses the optimal filter described here, using a hard-wired critical dose function. If new and improved critical dose measurements should become available it will be trivial to update this function. Additional information about Unblur is now included in the manuscript (section “Image processing”).